

# Spatial and Temporal Variability of Interhemispheric Transport Times

Xiaokang Wu[1,2], Huang Yang[1], Darryn W. Waugh[1], Clara Orbe[1,3], Simone Tilmes[4], and Jean-Francois Lamarque[4]

[1]Department of Earth and Planetary Sciences, Johns Hopkins University, Baltimore, Maryland, USA
[2]Department of Atmospheric Sciences, Texas A&M University, College Station, Texas, USA
[3]Goddard Earth Science Technology and Research, NASA Goddard Space Flight Center, Greenbelt, Maryland, USA
[4]National Center for Atmospheric Research, Boulder, Colorado, USA

*Correspondence to:* Darryn Waugh (waugh@jhu.edu)

**Abstract.** The seasonal and interannual variability of transport times from the northern mid-latitude surface into the southern hemisphere is examined using simulations of three idealized "age" tracers: A ideal age tracer that yields the mean transit time from northern mid-latitudes and two tracers with uniform 50-day and 5-day decay. For all tracers the largest seasonal and interannual variability occurs near the surface within the tropics, and is generally closely coupled to movement of the
intertropical convergence zones (ITCZ). There are, however, notable differences in variability between different tracers. The largest seasonal and interannual variability in the mean age is generally confined to latitudes spanning the ITCZ, with very weak variability in the southern extratropics. In contrast, for tracers subject to spatially uniform exponential loss the peak variability tends to be south of the ITCZ, and there is a smaller contrast between tropical and extratropical variability. These differences in variability occur because the distribution of transit time from northern mid-latitudes is very broad and tracers
with more rapid loss are more sensitive to changes in fast time scales than the mean age tracer. These simulations suggest that the seasonal/interannual variability in the southern extratropics of trace gases, with predominantly NH mid-latitude sources, may differ depending on the gases' chemical lifetimes.

## 1   Introduction

Interhemispheric transport is an important aspect for understanding the global distribution of tropospheric trace gases. In
particular, it is important to quantify the pathways and time scales for transport from northern hemisphere (NH) middle latitudes into the southern hemisphere (SH) as anthropogenic emissions of tropospheric ozone precursors, major greenhouse gases, aerosols, and ozone depleting substances occur primarily in the NH.

The majority of previous studies that have examined interhemispheric transport have used a simple two-box framework to quantify a single intermispheric exchange time, calculated in terms of the temporal change in the difference between the south-
ern and northern hemispherically integrated tracer mass (e.g., Levin and Hesshaimer, 1996; Geller et al., 1997; Lintner et al., 2004; Maiss et al., 1996; Denning et al., 1999). This metric is useful as it collapses all the transport into single parameter that can be used for model-data or inter-model comparisons. However, it is only a gross measure of interhemispheric transport,


with no information of spatial variations in transport times. In particular, it does not distinguish between transport into the southern tropics versus transport into the southern extratropics. Tracer observations and simulations support the existence of a strong tropical-extratropical transport barrier (e.g., Bowman and Carrie, 2002; Bowman, 2006; Miyazaki et al., 2008). In fact, Bowman and Carrie (2002) suggest that it may be more appropriate to use a three box model (with northern extratropical,
tropical, and southern extratropical boxes) to quantify tropospheric transport (see also Bowman and Erukhimova (2004)).

Alternatively, recent studies have used observed and simulated $SF_6$ or simulated idealized mean age tracers to estimate the mean transport time from the NH surface to locations throughout the troposphere (Holzer and Boer, 2001; Waugh et al., 2013). This approach provides a more complete description of interhemispheric transport, quantifying not only differences in transport into the tropics versus southern extratropics, but also differences in transport between the lower and upper troposphere.
However, these tracers (and two-box or three-box exchange times) only provide information about the mean transport time, whereas observations and models show there is a wide range of times and paths for transport from NH surface. More precisely, both observational-based estimates (Holzer and Waugh, 2015) and numerical simulations (Holzer and Boer, 2001; Orbe et al., 2016) of the transit time distribution (TTD) from the NH surface to the SH show very broad distributions, characterized by young modes and long tails. As a result, the mean transit time to SH locations, which controls the distributions of long-lived
trace gases, is much larger than the modal transit time, which is associated with the fast transport pathways that play a much more important role in controlling the distributions of chemical tracers with lifetimes of days to months.

Most of the focus in the above studies has been on the climatological mean transport, with only limited analysis of seasonal and interannual variability. This is especially the case for consideration of more than the mean transport time. For example, Orbe et al. (2016) performed pulse tracer releases at only four different times in a single year, which meant they could only do a
limited analysis of the seasonality and could not examine any interannual variability. Here we examine this issue, and examine the seasonality and interannual variability of transport from the NH surface into the SH, considering not only the mean transit times but also faster transport pathways and zonal variations in the transport.

The approach taken is to examine simulations of several tracers with the same NH source region but different time dependences (loss rates) (e.g., Waugh et al., 2003; Orbe et al., 2016). This approach does not enable the same detailed analysis of
TTDs as pulse release simulations (unless a large number of tracers are simulated), but does enable detailed analysis of seasonal and interannual variations (see next section for more discussion). Here we examine 30 year simulations of three idealized "age" tracers requested as part of IGAC/SPARC Chemistry-Climate Model Initiative (CCMI) (Eyring et al., 2013). One of the tracers (the NH clock or ideal age tracer) yields the mean transit time from the NH source region, while the other two tracers have 50 and 5 day loss rates and provide information on shorter transit times (that is more use for understanding the distributions of
short lived trace gases). The long simulations enable an examination of interannual, as well as seasonal, variations of transport into the SH.

The tracers and simulations examined are described in the next section, and the climatological distribution of the tracers presented in Section 3. Then the seasonal and interannual variations are examined in Sections 4 and 5, respectively, with concluding remarks in Section 6.





## 2   Methods

### 2.1   Tracers

We examine interhemispheric transport using simulations of three idealized age tracers: An ideal mean age tracer that yields the mean transit time from the NH source region, and two tracers with uniform decay time of 50 or 5 days.

The governing equation for the ideal mean age tracer $\Gamma(\mathbf{x}, t)$ is (Haine and Hall, 2002)

$$\frac{\partial \Gamma}{\partial t} + \mathcal{L}(\Gamma) = \Theta(t), \tag{1}$$

where $\mathcal{L}$ is the linear transport operator and $\Theta(t)$ is Heaviside function (zero for $t < 0$ and one for $t > 0$). The boundary condition is $\Gamma(\Omega, t) = 0$ where $\Omega$ is the source region, and $\Gamma(\mathbf{x}, 0) = 0$ initally. In other words, the tracer is initially set to a value of zero throughout the atmosphere, is held to be zero over $\Omega$, and subject to a constant aging of 1 year per year in the

rest of the model surface layer and throughout the atmosphere. Here $\Omega$ is the surface layer between 30°N to 50°N, and the ideal age tracer yields the mean transport time from this region. The ideal age tracer $\Gamma$ is referred to as age of air from northern hemisphere (AOA_NH) in CCMI (Eyring et al., 2013).

The two decay tracers have fixed concentration over $\Omega$ and undergo spatially uniform exponential loss, i.e.,

$$\frac{\partial \chi_T}{\partial t} + \mathcal{L}(\chi_T) = -\frac{1}{T}\chi_T \tag{2}$$

where $T$ is the constant decay time, $\chi_T$ is the concentration of tracer with decay time $T$, and $\chi_T(\Omega, t) = \chi_\Omega$ is a constant. We consider tracers with $T = 5$ and 50 days, that we referred to as the 5-day and 50-day loss tracers (the tracers correspond to NH_5 and NH_50 in CCMI).

In our analysis we express the concentration of the loss tracers as an "age"

$$\tau_T(r, t) = -T \ln\left(\frac{\chi_T(r, t)}{\chi_\Omega}\right). \tag{3}$$

This approach is common in oceanography (e.g., Waugh et al., 2003; Deleersnijder et al., 2001), and enables easier comparison with $\Gamma$. The basis for the age definition (3) can be seen by considering the idealized case of steady, advective flow with no mixing (i.e. $\mathcal{L}(\chi) = u \partial \chi / \partial r$, with $u$ a constant). The tracer concentration satisfying (2) is then given by $\chi_T(r, t) = \chi_\Omega \exp(-t_{adv}/T)$, where $t_{adv} = r/u$ is the advective time from the source region to the interior location, and equation (3) reduces to $\tau_T = t_{adv}$, i.e. for purely advective flow the tracer age (3) equals the advective time.

In the simple advective flow case the tracer age is independent of the tracer decay time $T$, and tracers with different decay rates yield the same age. However, this is not the case for more realistic flows with mixing, where the tracer age depends on the flow and the tracer decay $T$. For a steady flow with mixing the tracer age is (Waugh et al., 2003)

$$\tau_T(r) = -T \ln \int_0^\infty \mathcal{G}(r, t') e^{-t'/T} dt', \tag{4}$$

where $\mathcal{G}(r, t)$ is the distribution of transit times (referred to as the transit time distribution (TTD) or age spectra) from the source

region to $r$. Because of the exponential term inside the convolution integral in (4), tracers with different $T$ yield different $\tau_T$.





This is illustrated by considering a loss tracer with decay time $T$ is much larger than the width of the TTD, $\Delta$. In this case (4) reduces to (Hall and Plumb, 1994)

$$\tau_T \approx \Gamma - \Delta^2/T. \tag{5}$$

From this we can see that tracers with smaller $T$ have a younger $\tau_T$, and that for tracers with very slow decay the tracer age is close to the mean age ($\tau_T \to \Gamma$ as $T \to \infty$).

While the above means the tracer ages cannot be interpreted directly as a transport time scale, it does mean examination of tracers with different decay times highlight different aspects of the distribution of transit times (i.e. analysis of multiple tracers provides information on the characteristics of the TTD). Specifically, the age of a tracer is sensitive to the fraction of transit times less than the decay time of the tracer, but insensitive to transit time much longer than the decay time, as these long transit times carry very litte tracer mass.

## 2.2 Model and Analysis

The tracer fields examined here are from a simulation with the 4th version of the Community Atmospheric Model with troposphere-stratosphere chemistry (CAM4-chem) (Tilmes et al., 2015; Lamarque et al., 2012) run in "specified dynamics" mode using meteorology from Modern-Era Retrospective Analysis for Research and Application (MERRA) (Rienecker et al., 2011). This corresponds to the CAM4-REFC1SD simulation in Tilmes et al. (2016) and the CAM-C1SD simulation in the recent CCMI model intercomparison of Orbe et al. (2017b). Here we refer to the model simply as CAM.

The CAM simulation has horizontal resolution of 1.9° latitude by 2.5° longitude, 56 hybrid vertical levels from the surface to 1.87 hPa. For our analysis we interpolate from the hybrid levels to a standard set of isobaric levels spanning 1000 hPa to 10 hPa. The simulation examined was run from 1979 to 2010, after being "spun up" by running 5 years with 1979 meteorology. Here, we examine the monthly averaged fields from January 1980 to December 2009.

We examine the climatological seasonal-mean of the tracer ages (i.e. 30-year average for each month), as well as the seasonal and interannual variability. The seasonal variability is quantified by calculating the standard deviation of the climatological 12 month annual cycle, and is referred to as $\sigma_\tau^{\mathrm{seas}}$ (with $\tau = \Gamma$, $\tau_{50}$, or $\tau_5$). The interannual variability is similarly quantified by calculating the standard deviation over 30 years. To minimize the impact of seasonality, the interannual variability is calculated for each season, i.e., $\tau$ is averaged over every three months and the standard deviation is calculated of these seasonal means. We focus here on interannual variability for December to February (DJF) and June to August (JJA), which we refer to as $\sigma_\tau^{\mathrm{DJF}}$ and $\sigma_\tau^{\mathrm{JJA}}$. (For both seasonal and interannual variability the calculations of the standard deviation are performed at individual locations, and any zonal averaging is done after these calculations.)

## 3 Climatological Distributions

We first examine the climatological seasonal-mean distributions of the tracer ages. Fig. 1 shows the zonally averaged $\Gamma$, $\tau_{50}$, and $\tau_5$ for northern winter (DJF) and summer (JJA). There is a similar distribution for the different tracer ages, with the smallest





values in northern mid-latitudes (close to the source region), oldest surface values at the south pole, weak meridional gradients in northern extratropics, largest meridional gradients in tropics, and relative weak vertical gradients at all latitudes (with slightly positive vertical gradients in the northern hemisphere (NH) and slightly negative gradients in the southern hemisphere (SH)). The spatial distribution of the tracer age shown in Fig. 1 are similar to the distribution of idealized or realistic long-lived

tracers shown in previous studies (e.g., Denning et al., 1999; Holzer and Boer, 2001; Miyazaki et al., 2008; Waugh et al., 2013), and can be related to differences in meteorology and transport between regions. There is rapid transport from the NH mid-latitude surface into the NH extratropical troposphere, through a combination of along-isentropic and convective mixing, and as a consequence there are weak age gradients in the NH extratropics. There is also rapid low-level transport from NH mid-latitudes into the tropics, but the transport into the SH is "slowed" by convection and rapid vertical mixing associated with the

intertropical convergence zone (ITCZ), resulting in large surface meridional age gradients near the ITCZ. The rapid vertical mixing within tropical convection results in very weak vertical tracer gradients within the tropics, and the strong meridional gradients in the tropics persistent into the middle troposphere. In the tropical upper troposphere there is increased meridional transport due to the upper branch of the Hadley Cell, and this results in weaker meridional tracer gradients.

While there is qualitative agreement in the spatial distributions of the different tracer ages, there are substantial quantitative

differences. First, there are large differences in the magnitude of the ages, especially in the SH where $\Gamma \gg \tau_{50} \gg \tau_5$ (consistent with equation (5)). Second, there are differences in the meridional gradients: the meridional gradients of $\Gamma$ in the tropics are much larger than those in the SH (where $\Gamma$ is nearly constant), whereas the meridional gradients of $\tau_5$ are similar in the tropics and SH. These differences are illustrated in Fig. 3(a,b) which shows the latitudinal variation of the tracer ages at 900 hPa, for DJF and JJA.

These quantitative differences among the tracer ages occur because the TTDs in the tropics and SH are very broad (Holzer and Waugh, 2015; Orbe et al., 2016), and the tracers are sensitive to different aspects of the TTDs. As discussed above, $\tau_5$ is most sensitive to the shorter transit times whereas $\Gamma$ is the mean of the TTD and is dependent on the long tail of old transit times. The differences in meridional gradients of the two ages are related to changes in the shape of the TTD with latitude. Orbe et al. (2016) showed there is a transition in shape of the TTD from north of the ITCZ to south of the ITCZ, which they attributed to a change

in the relative contribution of rapid, advective pathways from northern mid-latitudes and slow eddy diffusive recirculation of "old" air into the tropics from the SH. The latter has a much larger impact on $\Gamma$ than on $\tau_5$, resulting in a much larger increase in $\Gamma$ across the ITCZ but relatively constant values in the southern extratropics. By comparison, $\tau_5$ is most sensitive to very short transit times as it is determined more by rapid advective pathways, resulting in roughly constant meridional gradients of $\tau_5$ throughout the SH.

The latitudinal gradients in the tracers are much larger than zonal gradients, but there are still some zonal variations. This is illustrated in Fig. 2 which shows the 900 hPa distribution of the climatological $\Gamma$ and $\tau_5$ for (a,c) DJF and (b,d) JJA. There are weak zonal variations in the extratropics for both tracers, but noticable zonal variations within the tropics. For example, in DJF the mean age over the equatorial Indian Ocean is smaller than over the equator of other oceans, whereas in JJA the mean over the northern tropical Indian Ocean is larger than over the Pacific or Atlantic oceans.





## 4    Seasonal Variability

Comparison of the left and right panels of Fig 1 shows seasonal differences in the tracer age distributions, which are again qualitatively similar among the tracers. For example, the location of the largest surface meridional gradients are south of the equator during DJF but north of the equator during JJA, and the near-surface tracer ages at the equator and in the southern tropics are younger in DJF than in JJA (Fig. 3(a,b)). There are also seasonal differences away from the surface, with older ages in DJF than in JJA in both northern and southern subtropical middle-upper troposphere.

These seasonal differences in the tracer ages are linked to the seasonally-varying Hadley circulation (e.g., Bowman and Cohen, 1997; Bowman and Erukhimova, 2004). The largest surface age gradients occur at the ITCZ, with young ages north of the ITCZ and older ages south. The latitude of the ITCZ moves with season and there is a corresponding north-south shift in the latitude of large meridional age gradients, i.e. largest surface meridional gradients are south of the equator during DJF but north of the equator during JJA (Fig. 1, 3(a,b)). This results, as will be shown below, in a large seasonality at locations within the seasonal range of the ITCZ, with older ages when the ITCZ is north of the location and younger ages when it is to the south.

The seasonality of the Hadley circulation can also explain the seasonality in the tracers in the northern subtropical middle troposphere and southern tropical upper troposphere. During DJF the northern cell is strongest (see arrows in Fig. 1a) and transports "older" ages from the equatorial upper troposphere into the northern subtropical middle troposphere (resulting in older ages in DJF than JJA), whereas during JJA the stronger southern cell (Fig. 1b) increases the transport of "young" air into the southern subtropical upper troposphere (again resulting in older ages in DJF than JJA).

As with the climatological distributions, there are quantitative differences in the seasonality (DJF-JJA differences) of the different tracer ages. In particular, the seasonality of near-surface $\Gamma$ south of $20°$S is much smaller than at the equator, whereas for near-surface $\tau_5$ there is a smaller decrease in the seasonality from the equator to southern mid-latitudes. This can be seen clearly in Fig. 3(c) which shows the latitudinal variation in the seasonal standard deviation $\sigma_\tau^{\text{seas}}$.

As mentioned in the previous sections there are zonal variations in the tracer ages within the tropics, that vary with season. These zonal variations in the ages are consistent with variations in the ITCZ, see surface winds (arrows) and convergence (contours) in Fig. 2. The ITCZ is close to the equator in both DJF and JJA over the Pacific, whereas there is a large seasonal variation of ITCZ over the Indian ocean: it is well north of the equator during JJA but south of the equator in DJF. Similar variations occur for the regions of largest meridional age gradients. Associated with the seasonal movement of the ITCZ there is a change in direction of the surface winds, with the largest changes again occurring in the Indian ocean sector. In particular, in the tropical western Indian ocean there is a strong southward flow during DJF, but a strong northward flow in JJA. This seasonality in wind direction results in a large seasonality in the age.

The spatial variation of the seasonality, and differences between $\Gamma$ and $\tau_5$, can be seen clearly in Figs. 4 and 5 which show maps of surface and vertical cross-sections, respectively, of the seasonal standard deviation $\sigma_\tau^{\text{seas}}$. Consistent with the above discussion, the largest values of both $\sigma_\Gamma^{\text{seas}}$ and $\sigma_{\tau_5}^{\text{seas}}$ are within the tropics. However, while the peak $\sigma_\Gamma^{\text{seas}}$ and $\sigma_{\tau_5}^{\text{seas}}$ occur at similar latitudes over the Pacific and Atlantic Oceans, the peak $\sigma_{\tau_5}^{\text{seas}}$ is south of the peak $\sigma_\Gamma^{\text{seas}}$ in the Indian Ocean sector. Also, $\sigma_\Gamma^{\text{seas}}$ in the southern extratropics is much smaller than in the tropics ($\sigma_\Gamma^{\text{seas}}$ is as high as 180 days in the tropics but only



round 10 days in the southern extratropics), where $\sigma_{\tau_5}^{\mathrm{seas}}$ is comparable in the tropics and southern extratropics (5-10 days). Fig. 5 also shows that large seasonality is generally only near the surface (pressures above 800 hPa). However, there is moderate to larger seasonality in the northern subtropical mid-troposphere, southern subtropical upper troposphere over the Indian ocean, and near the tropopause (especially for $\tau_5$).

The seasonal movement of the ITCZ can explain much of the seasonality in the tracer ages. In particular, the north-south movement of the ITCZ results in a similar movement of the region of high meridional age gradients, and large seasonality of tracer age for tropical locations, i.e., as the ITCZ moves from north to south of a particular location there will be decrease in age, and vice-versa for a northward shift. The seasonal migration of the ITCZ varies with longitude, with a much larger variation over the Indian ocean than over the Eastern Pacific (e.g., Waliser and Gautier, 1993; Gloor et al., 2007). This is shown by

contours in Figs. 2 and 4. This results in a wider range of latitudes that the ITCZ crosses during the annual cycle, and hence larger seasonality in tracer ages over the Indian ocean than over the Eastern Pacific (Fig. 4a). There are some differences in locations of peak seasonality of $\Gamma$ and $\tau_5$ over the Indian Ocean sector, with the peak $\sigma_{\Gamma}^{\mathrm{seas}}$ occurring north of the equator while the peak $\sigma_{\tau_5}^{\mathrm{seas}}$ is at or south of the equator. These differences are again related to differences in the mean meridional gradients of the tracers: Only in regions with large meridional gradients do perturbations of the circulation lead to large changes in the

tracer ages.

To quantify the age-ITCZ relationship we compare the latitudinal movement of the ITCZ with the age at a fixed location. Fig. 6 shows the relationship between the simulated 900 hPa $\Gamma$ or $\tau_5$ with the ITCZ latitude (calculated as the latitude of maximum convergence at 900 hPa between 15°S-30°N for each longitude) for four different longitudes (corresponding to the Indian, Western Pacific, Eastern Pacific or Atlantic oceans). For both tracer ages and all locations there is a positive correlation,

i.e. older age for a more northern location of the ITCZ. There are some differences in the age-ITCZ relationships between the ocean basins, with a more compact, linear relationship over the Atlantic and Eastern Pacific than other regions. Over the Indian Ocean the age-ITCZ relationship is nonlinear, especially for $\Gamma$, with a more rapid change of age with latitude of the ITCZ when the ITCZ is south of 10°N than north.

Observational evidence for the above relationship between the seasonality of tracer ages and latitude of the ITCZ is found

in the estimates of "$SF_6$ age" from surface measurements of $SF_6$. Waugh et al. (2013) showed that there are large annual cycles of $SF_6$ age dervied from measurements in the tropical Indian (Mahe Island, Seychilles; 4.7S, 55.5E ) and Eastern Pacific (Christmas Island; 1.7N, 157.1W ) oceans. The variation of the $SF_6$ age with latitude of ITCZ at these stations (Fig. 7) is similar to those for $\Gamma$ shown in Fig. 6a,c, including the linear relationship for the Eastern Pacific station but nonlinear relationship for the Indian Ocean station.

**5  Interannual Variability**

We now examine the interannual variability of the tracers, first for northern winter (DJF) and then northern summer (JJA).





## 5.1 Northern Winter

As for seasonal variations, the interannual variations of the DJF ages ($\sigma_\tau^{\mathrm{DJF}}$) are largest in the tropics - subtropics, and the regions of largest variability for $\tau_5$ are south of those for $\Gamma$ (Figure 8). The interannual variability is, however, weaker than the seasonality, e.g. the maximum $\sigma_\Gamma^{\mathrm{DJF}}$ is around 50 days compared to 180 days for $\sigma_\Gamma^{\mathrm{seas}}$. There are also differences in the locations of peak seasonal and interannual variability. For example, the peak $\sigma_\Gamma^{\mathrm{DJF}}$ over the Indian sector is in the central equatorial Indian Ocean, whereas the peak $\sigma_\Gamma^{\mathrm{seas}}$ is north of the equator (with two local maximum). A similar difference in locations of peak values occurs between $\sigma_{\tau_5}^{\mathrm{DJF}}$ and $\sigma_{\tau_5}^{\mathrm{seas}}$, and the tropical - extratropical difference in $\sigma_{\tau_5}^{\mathrm{DJF}}$ is much smaller than that for $\sigma_{\tau_5}^{\mathrm{seas}}$ (see also Fig. 3). Again consistent with seasonal variability, the interannual variability is largest near the surface and generally small in the upper troposphere (not shown). The regions of highest $\sigma_\Gamma^{\mathrm{DJF}}$ are generally close to the location of the ITCZ or the South Pacific convergence zone (SPCZ), suggesting that the interannual variability of $\Gamma$ is again connected to variability in the surface convergence and to the location of the strongest mean tracer gradients.

Several previous studies have linked variability in interhemispheric transport to the El Nino - Southern Oscillation (ENSO) (e.g, Elkins et al., 1993; Prinn et al., 1992; Lintner et al., 2004; Waugh et al., 2013). We examine this relationship by calculating the correlation between the 30-year times series of DJF $\Gamma$ at each location with the Ocean Nino Index (ONI) (where ONI $>$ 0.5 indicates an El Nino event, while ONI $<$ -0.5 indicates a La Nina event). A 30-year time series is too short to do a detailed analysis of ENSO and transport, but it does provide some guidance on possible ENSO related variability. As shown in Fig. 9a there are coherent regions with large positive or negative $\Gamma$-ONI correlations, with both occuring either side of the equator. There is large region with positive correlation in the southern subtropical central Pacific (near 170°E), but negative correlations are found in southern subtropical eastern Pacific and Indian oceans. Thus, during an El Nino year there tends to be older ages over the southern subtropical central Pacific but younger ages over the southern subtropical eastern Pacific and Indian oceans, and the reverse for La Nina years. The age-ONI correlations at and north of the equator are generally the opposite sign to those south of the equator at the same longitude, i.e. there are negative correlations in northern tropical central Pacific. A similar pattern of correlations with ENSO is also found for $\tau_5$, with region of positive correlations in south Pacific slightly south of that for correlation with $\Gamma$ (not shown).

The above age-ENSO correlations are illustrated in Fig. 9b-e which shows maps of DJF $\Gamma$, CAM precipitation (as a proxy for the intensity of tropical convection), and surface winds for a strong El Nino year (1998, ONI=2.1) and a strong La Nina year (2000, ONI=-1.6); in panels (b,c) the full fields are shown whereas in panels (d,e) the anomalies from the 30-yr climatology are shown. There is a large difference in precipitation over the Pacific between these years: During the El Nino year there is high precipitation located south of the equator around 5°S that extend across the Pacific, while in the La Nina year the precipitation is higher on western than eastern side of the Pacific and there are two regions of high precipitation fluxes, one north ($\sim$10°N) and the other south ($\sim$ 15°S) of the equator. These difference in the location of peak convection result in differences in transport to the equatorial western-central Pacific. During the El Nino year there is rapid, direct low-level transport to the equator as the deep convection is south of the equator, consistent with younger ages. In contrast for the La Nina year the convection around 10°N reduces this direct transport and the $\Gamma$ is older in the same region (consistent with the negative correlation shown in Fig.





9a). The reverse correlation occurs south of the equator because of ENSO variations in deep convection in the SPCZ region, which modify the transport of very old air from southern extratropics back into the southern tropics, i.e., during La Nina years the SPCZ convection reduces this transport resulting in younger ages in southern tropical Pacific.

The sign of the age-ENSO correlations over the eastern Pacific - Atlantic and the Indian oceans are opposite to that over the western-central Pacific, i.e. there is positive correlation in the northern tropics over eastern Pacific but a negative correlation over the western Pacific (Fig. 9a). The cause of this is not clear, although it likely due to El Nino - La Nina differences in the subtropical surface flow over these these regions. For example, during the 1997-1998 El Nino year the equatorial winds over the equatorial Indian ocean have a stronger than average northward component, which transport more old, southern hemisphere, air across the equator resulting in older age in northern tropical Indian ocean.

Obervational support for the above age-ENSO correlation is found in trace gas measurements at America Samoa (14°S, 170°W). Measurements of methyl chloroform and CFCs from this station show lower concentrations (indicating slower transport from NH sources) during El Nino year (e.g, Elkins et al., 1993; Prinn et al., 1992). As America Samoa lies just inside the region of positive age-ONI correlation, this is consistent with the above simulated variability. The simulations indicate that the observed result of slower transport to the SH during El Nino years may hold only in the western-central Pacific, and there could be faster transport to the eastern Pacific or Indian subtropical oceans. Unfortunately similar multi-year trace gas measurements are not available from these locations to test this.

Some caution is needed with the simulated age-ENSO relationship as it is based only on a 30 year simulation, that includes only two major El Ninos (1982/83 and 1997/98). However, preliminary analysis of the age-ENSO relationship in a CAM4chem REFC2 simulation covering 1960 to 2100 yields correlation patterns similar to those shown in Fig. 9a (not shown), including high negative and positive correlations either side of the equator in western-central Pacific and the opposite signed correlations over the eastern Pacific and Indian Oceans.

## 5.2 Northern Summer

The general characteristics of the interannual variability during northern summer (JJA) is similar to that in winter, i.e., the largest variability is in the tropics and there is small variability in the SH (especially for $\Gamma$), see Figure 10a,b. The location of the peak interannual variability at the surface varies between seasons, with the peak in $\sigma_\tau^{\mathrm{JJA}}$ generally north of that for $\sigma_\tau^{\mathrm{DJF}}$. This is consistent with the more northern location of the ITCZ in JJA, i.e. the peak standard deviation for each season is close to the climatological location of the ITCZ for that season. As in DJF, the largest $\sigma_{\tau_5}^{\mathrm{JJA}}$ is located south of peak $\sigma_{\tau_5}^{\mathrm{JJA}}$. This is especially true in the Indian Ocean sector, where $\sigma_\Gamma^{\mathrm{JJA}}$ is largest around 20°N but $\sigma_{\tau_5}^{\mathrm{JJA}}$ is largest around 5°S. This difference between the tracer ages is again consistent with the differences in their meridional gradients, e.g. there are weak $\Gamma$ gradients in the tropical Indian ocean but large $\tau_5$ gradients in southern tropics.

While the variability of the tracer ages generally decreases with height from the surface, this is not the case for $\sigma_\Gamma^{\mathrm{JJA}}$ over the western tropical Indian ocean. Here there is very little interannual variability near the surface for this region, but as shown in Fig. 11 there is a region of high interannual variability at 650 hPa that extends from tropical Africa over the Indian Ocean. Near the surface the largest $\sigma_\Gamma^{\mathrm{JJA}}$ occurs around 30°N, but around 650 hPa the largest variability is around 5°N. The large $\sigma_\Gamma^{\mathrm{JJA}}$



near the surface can be attributed to the variations of ITCZ (surface convergence), but variability in the ITCZ does not account for the large variability near 650 hPa. A possible cause of the large $\sigma_\Gamma^{\text{JJA}}$ at 650 hPa is variability in the ascent in the lower-mid troposphere over this region. During JJA there is a narrow region of strong ascent in the lower-mid troposphere over tropical Africa - Indian Ocean, between the African easterly jet and tropical easterly jet, that does not extend down to the surface but

does produce large precipitation in a "tropical rainbelt" south of the surface ITCZ (e.g., Nicholson (2009)). This region of strong ascent likely impacts meridional tracer transport, and the large $\sigma_\Gamma^{\text{JJA}}$ at 650 hPa could be connected to variability in ascent. This possibility requires further examination.

## 6  Conclusions

The seasonal and interannual variability of transport times from the northern hemisphere mid-latitude surface into the tropics
and southern hemisphere has been examined using simulations of idealized "age" tracers. For all tracers the largest seasonal and interannual variability occurs near the surface within the tropics, and is generally closely coupled to variability in the tropical convergence zones (ITCZ, SPCZ). The seasonal migration of the ITCZ is responsible for the majority of seasonality in the tracer ages (with younger ages when the ITCZ is further south), while a large amount of the interannual variability during DJF is due to ENSO-related variations in surface convergence and convection, especially over the Pacific Ocean.

There are, however, notable differences in the variability of tracers with different time dependencies. The largest variability in the mean age ($\Gamma$) is confined to the tropics, generally close to the location of the ITCZ (or SPCZ), and there is very weak seasonal or interannual variability in the southern extratropics (e.g., the interannual standard deviation of $\Gamma$ in the southern extratropics is less than 1% of the climatological mean value). In contrast, for the 5-day and 50-day loss tracers the peak variability of the age of tracer tends to be south of the ITCZ, and there is a smaller contrast between tropical and extratropical
variability. For example, the DJF interannual standard deviation the age of the 5-day loss tracer ($\tau_5$) is around 30-40% of the mean in both the tropics and southern mid-latitudes.

These differences in temporal variability of the tracers occur because the tracers are sensitive to different aspects of the TTD (e.g., $\tau_5$ is more sensitive to changes in the fast transit scales than $\Gamma$), and this results in differing meridional age gradients. Orbe et al. (2016) noted that fast (advective) transport pathways make only a very small contribution to the TTD south of the
ITCZ, and the TTD is dominated by slow (eddy-diffusive) pathways. Changes in these fast transport pathways south of the ITCZ can cause substantial variations in tracers with rapid loss (e.g., $\tau_5$) as these tracers are sensitive to changes in the fast time scales (and insensitive to changes in transit times much longer than a month as these carry little tracer), but have much weaker impact of the mean transit time (with more strongly influenced by the trail of the TTD).

One possible concern with this analysis is that it considered only one model. However, analysis of the same tracers from
several other models (those considered in (Orbe et al., 2017a)) yield similar patterns of seasonal and interannual variability, and connections to the ITCZ and SPCZ, as presented here (not shown). This suggests that the results presented here are robust, but this will need to be examined using the full suite of models participating in CCMI. An open question is the magnitude of longer term variability and trends in the transport times. This analysis here suggests this will likely be small, but this needs to





be examined. The CCMI model simulations of the 21st century offer an opportunity to perform such analysis, and to also test the robustness of the age-ENSO variability presented here.

The differing seasonal and interannual variability of the idealized tracers suggest that the seasonal/interannual variability in the southern extratropics of trace gases, with predominantly NH sources, may differ depending on the chemical lifetimes of the

5   gases. For tracers with very long lifetimes (e.g., $SF_6$ and CFCs) we may expect very weak temporal variability due to transport, whereas for tracers with shorter lifetimes (e.g. non-methane hydrocarbons) there may be noticeable transport-induced seasonal or interannual variability. Conversely, our study also suggests that combinations of tracers with different lifetimes may be used to constrain the TTD from observations. This possibility requires further examination.

*Acknowledgements.* This work was supported by NSF Grant AGS-1403676 and NASA Grant NNX14AP58G. The CAM4chem output can

10   be downloaded from https://www.eathsystemgrid.org/search.html?freeText=ccmi1 and the ONI data from http://www.cpc.ncep.noaa.gov/.



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





**Figure 1.** Latitude-pressure distribution of the climalogical seasonal-mean zonal-mean (a,b) $\Gamma$ (c,d) $\tau_{50}$, and (e,f) $\tau_5$, for boreal winter (DJF) and boreal summer (JJA). Arrows show the meridional circulation, thin contours show isentropes (contours every 20 K), and the thick contour shows the tropopause.





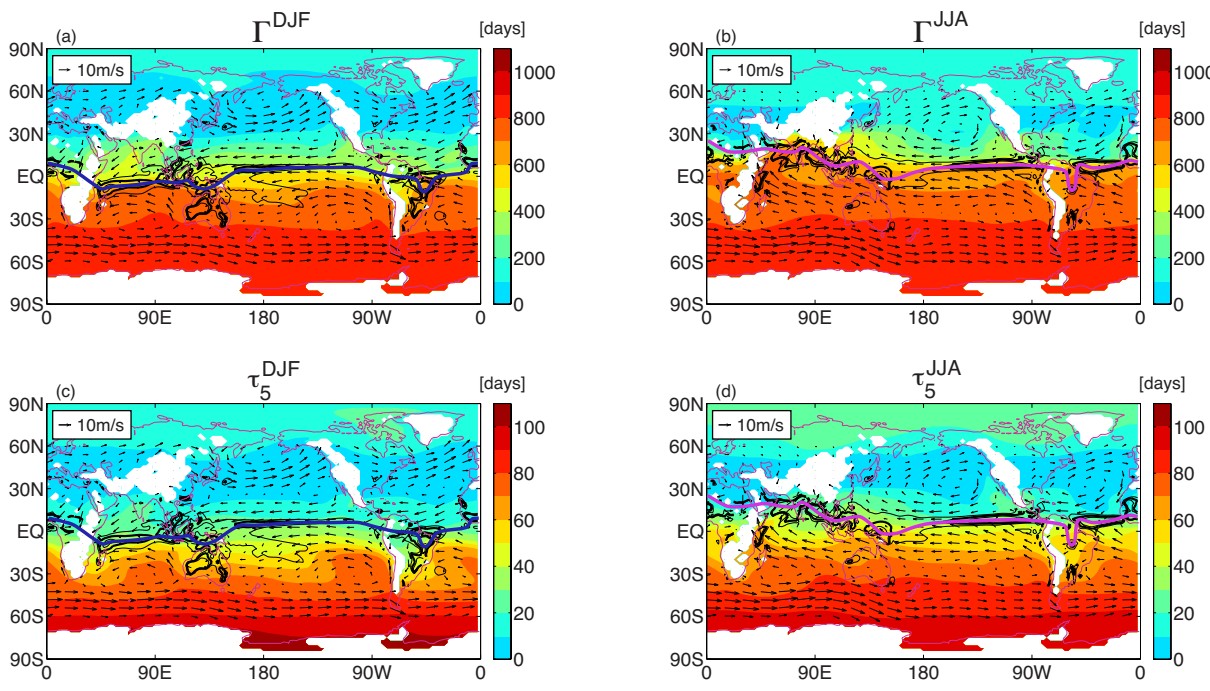

**Figure 2.** Climatological-mean $\Gamma$ and $\tau_5$ at 900 hPa for (a,c) DJF and (b,d) JJA. Arrows show horizontal velocity ,black contours convergence at 900 hPa (contours at (-3,-2,-1) $\times 10^6$ s$^{-1}$, with -2 $\times 10^6$ s$^{-1}$ bold), and blue/pink curves approximation location of the ITCZ.





**Figure 3.** Latitudinal variation of (a) DJF climatological mean, (b) JJA climatological mean, (c) seasonal standard deviation, (d) DJF interannual standard deviation, and (e) JJA interannual standard deviation for $\Gamma$ $\tau_{50}$, and $\tau_5$ at 900 hPa. For all panels the quantity for $\tau_{50}$ is multiplied by 3 and that for $\tau_5$ multiplied by 8.



**Figure 4.** Maps of seasonal variatiability ($\sigma_S$) of (a) $\Gamma$ and (b) $\tau_5$ at 900 hPa. Thin contours show climatological mean tracer ages (in days), and blue and pink curves approximation location of the ITCZ in DJF and JJA, respectively.





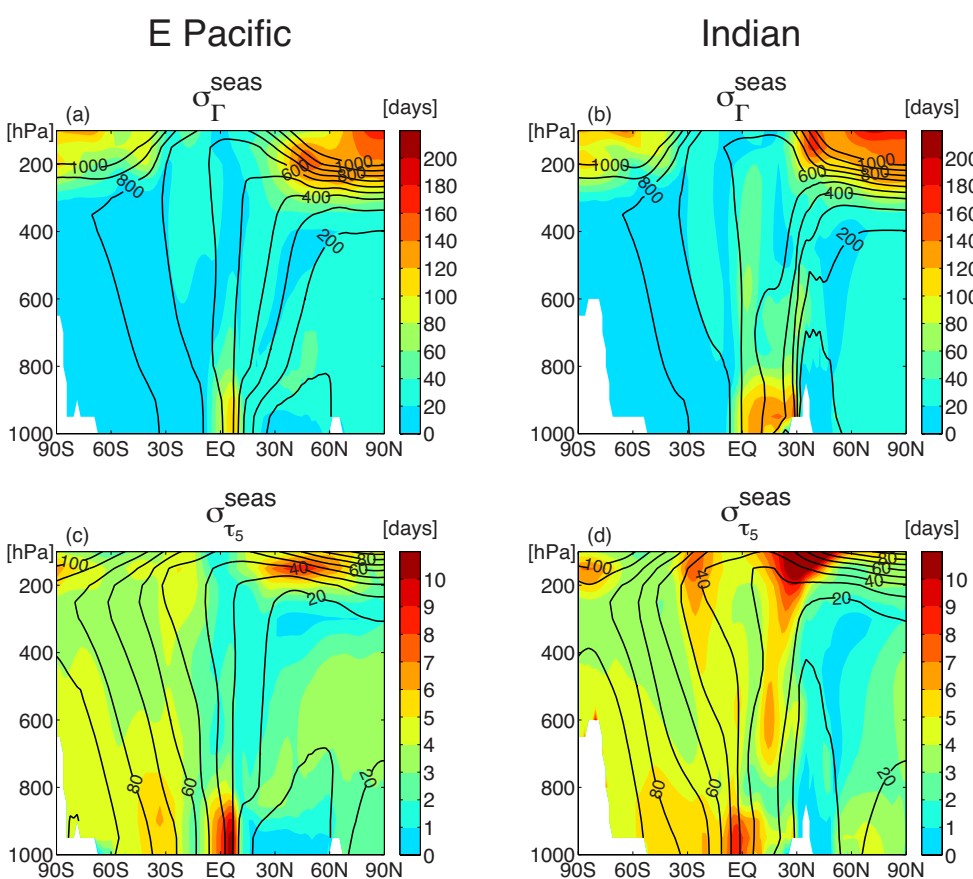

**Figure 5.** Latitude-pressure variation of ($\sigma_S$) of (a,b) $\Gamma$ and (c,d) $\tau_5$, for Eastern Pacific (150-120°W) and Indian Ocean (60-90°E) sections. Contours show climatological mean distributions of (a,b) $\Gamma$ and (c,d) $\tau_5$ (in days).



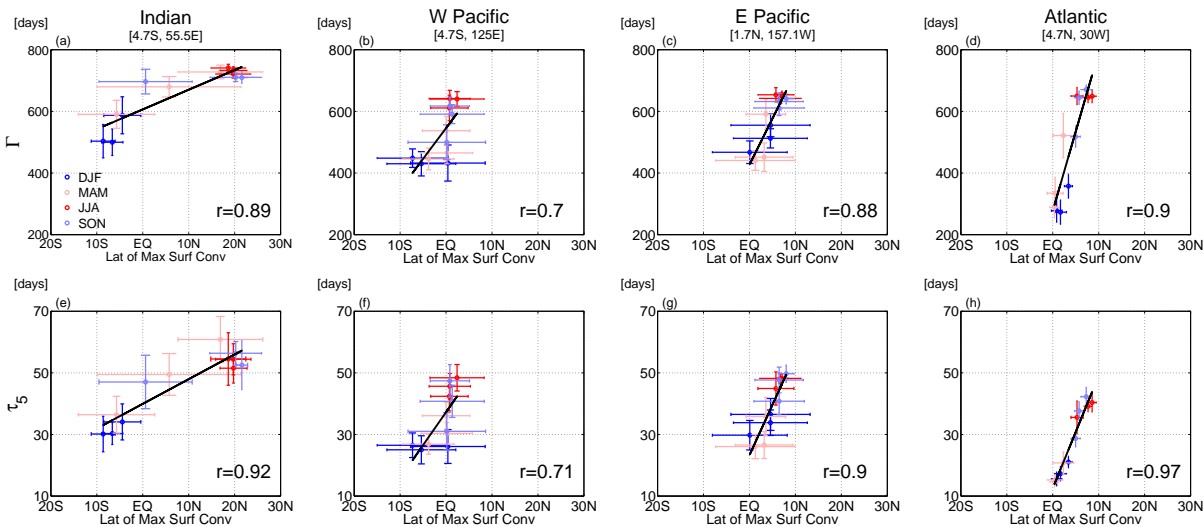

**Figure 6.** Relationship between the latitude of maximum surface convergence and tracer age at 900 Pa for locations in the (a,e) Indian, (b,f) West Pacific, (c,g) East Pacific, and (d,h) Atlantic oceans. Top row shows $\Gamma$ and bottom row $\tau_5$. Coordinates of the locations are shown above panels (a)-(d). Width of the horizontal or vertical bars are twice the interannual standard deviation, and different colors represent differe seasons (see panel a). Black line shows linear fit, and correlation coefficient is given within each plot.




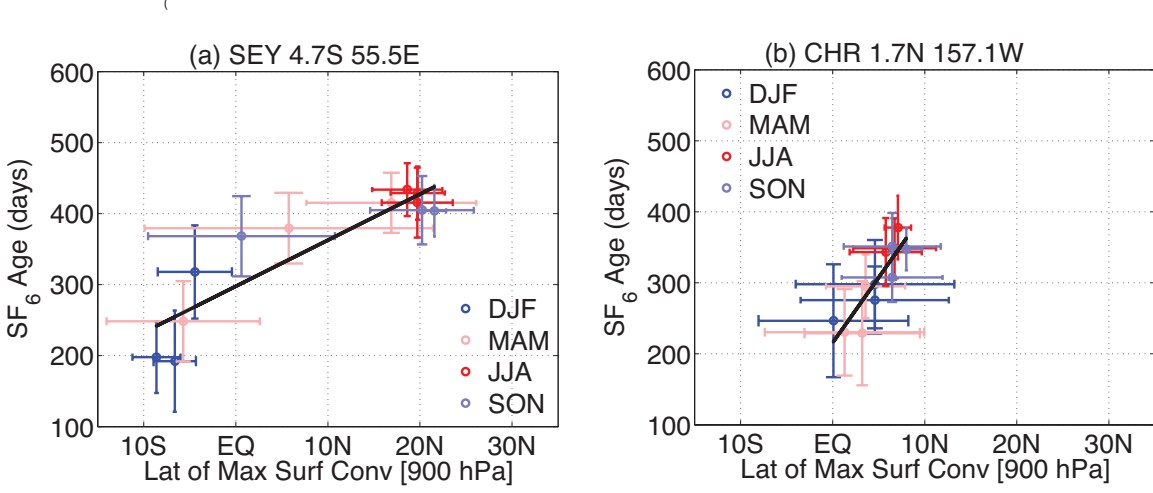

**Figure 7.** As in Figure 6a and c except for the relationship between observed SF$_6$ age with the latitude of the ITCZ, for measurements from (a) Seychilles and (b) Christmas Island.





**Figure 8.** As in Figure 4 except for DJF Interannual standard deviation.




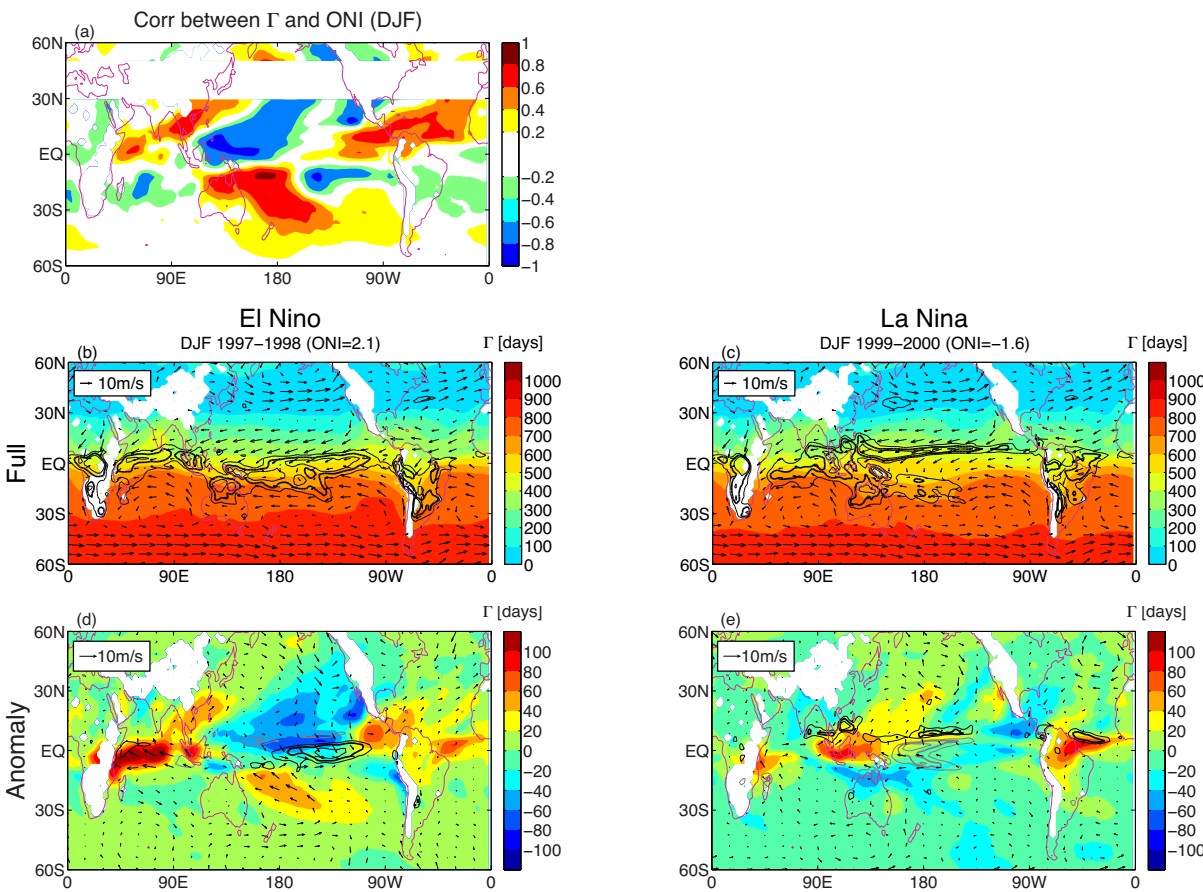

**Figure 9.** (a) Correlation betwen ONI and $\Gamma$ at 900 hPa. Correlations with absolute value larger than 0.361 are significant at 95% confidence level. (b,c) $\Gamma$ (shading), 900 hPa horizonal winds (arrows) and precipitation (contours; (4,6,8) mm/day, with the 6 mm/day bold) for (b) DJF 1997-98 (El Nino winter) and (c) DJF 1999-2000 (La Nina winter). (d,e) As in (b,c) each anomalies from climalogical DJF fields. In (d) black contours show precipitation anomalies of (4,6,8) mm/day, and gray contours show precipitation anomalies of (-8,-6,-4) mm/day; while in (e) black contours show precipitation anomalies of (2,3,4) mm/day, and gray contours show precipitation anomalies of (-4,-3,-2) mm/day.





**Figure 10.** Same as 8 except for JJA Interannual standard deviation.





**Figure 11.** Same as 10 except for 650 hPa.