# Peer review of "Spatial and Temporal Variability of Interhemispheric Transport Times"

_Atmospheric Chemistry and Physics, 2017_

## Referee Comment (RC1) · Anonymous Referee #1 · 31 Jan 2018

Review of "Spatial and Temporal Variability of Interhemispheric Transport Times"

In this paper, the authors investigate the transport of air from the northern hemisphere (NH) to the southern hemisphere (SH), including the seasonal and interannual variability of this transport, using a specified dynamics model. Three different tracers are utilized in order to highlight different aspects of the distribution of transit times without calculating the transit time distributions explicitly. The results–that the seasonal variability in the tracer transport is driven by the movement of the ITCZ and that the interannual variability is linked with ENSO–are unsurprising and consistent with Orbe et al. 2016 and Waugh et al. 2013. The authors have done several new things in this study, and the paper would be improved by highlighting the novel aspects (interannual variability, robust seasonal analysis, comparison with data) of the work. Generally,

more discussion of mechanisms, why the results matter, and improved framing would take this paper from being a simple advance in the examination of inter-hemispheric transport to being a more interesting and useful contribution. Overall, this paper currently provides a clear explanation of transport timescales from NH midlatitudes and explains the difference in using tracers with different losses. It would be much stronger if these results were placed into chemical and dynamical context. Because of the extent of these changes, I have recommended the paper be accepted after major revisions, although depending on how much the authors have already thought about these concerns, it's possible the revisions could be very quick.

I have provided specific areas for improvements below, with both specific and general suggestions.

Introduction: The previous literature on interhemispheric transport is treated well. It is not clear from the discussion why neglecting the seasonal and interannual variability in inter-hemispheric transport matters, and why it is important to consider both shorter lived and longer lived trace gases. There are a lot of potential ways to do this–perhaps discuss with respect to the availability of OH and its seasonal distribution (e.g. Lelieveld et al. 2016)? Or look into the work of Prather and Holmes (2013) and see if their work on lifetimes provides a complementary approach or useful motivation. Or what about $CO_2$ uptake in the SH? A stronger connection to the chemistry would go a long way for motivating the rest of the study.

In each section, explain what we expect to see. The ITCZ is known to drive interhemispheric transport and ENSO is known to change tropical variability in DJF, so the results are not surprising. Unfortunately the paper currently reads as a methodical discussion of plots. In my opinion, the story of this paper would be more compelling if it were presented as "this is what we expect to see", "let's check–this is what we see", and "these are interesting details". For the ENSO discussion, an explanation of why the weaker Walker circulation would lead to the pattern of age difference would be nice. The comparison with the wind anomalies in Wang and Fiedler (2006) in Fig. 2c definitely makes

Fig. 9a seem reasonable. The current discussion based on the case study comparison of one El Nino and one La Nina year is less convincing than a more general discussion would be. For example, if the variations in deep convection in the SPCZ region are the norm for La Nina years, a citation here would be helpful. For JJA interannual variability, it is not as obvious what mechanisms are at play. Perhaps the variability is related to changes in the monsoon and therefore the phasing of the MJO (the strong variability in age and the mean age contours are both apparently coincident with the Somali jet, and the authors discuss the ascent in this region). Since ENSO is the primary signal of interannual climate variability, I would also think that checking whether JJA age is correlated with the ONI would be worthwhile, just in case. I do not expect the authors to do any extensive calculations for these dynamical connections—rather, I ask for a discussion of the transport in the context of the tropical dynamics. I think a different, longer model run would be necessary to get at the details here, and that is beyond the scope of this paper.

I find the comparison to observations makes this paper more valuable; this should be emphasized in the introduction, and the discussion should be expanded (p. 7).

In the conclusions, make the point that by verifying these results are consistent with expectations, observations, and previous modeling studies that use TTDs, this study has demonstrated how useful these tracer diagnostics are for understanding transport. The clear explanation of the sensitivity of different tracers to different parts of the TTD and the exploration of the variability of transport from this paper is very valuable, even if much of this has been implied in Orbe et al. 2016, 2017. The point that is made in this conclusion is that by examining a bunch of CCMI models, the robustness of these relationships could be tested. I would contend that we fully expect the gradients of age to be tightly coupled to convection and that Orbe's work has shown as much–these relationships don't really need to be tested. However, differences in these relationships between models might have interesting implications for the impacts of dynamics and convective parameterization on model transport and chemistry and how they differ in

the CCMI models.

Specific edits (content ones have a *, otherwise these are mostly grammar):

P1L2: "A"–> "an"

P1L7: comma after "loss"

P1L11: no comma after "gases"

P1L14: "aspect for" strange wording–aspect of? or just "is important for"?

P1L19: "intermispheric"–> "interhemispheric"

P2L20-1: Here we examine this issue and examine the seasonality . . .–> Here we examine the seasonality . . .

P3L3: no "mean"

P3L5: no "mean"

P3L9: insert "is" between and and subject

P2L24: comma would be better as semicolon

P2L27: comma after mixing

P2L29: number disagreement between "distribution" and "spectra"–I think you mean spectrum.

P4L1: with ˆa decay time T ˆthat is . . .

P4L6: "the above" is ambiguous–the above what?

*P4L16: Why do you use CAM SD? What advantages (and disadvantages) does using the nudged run have for this study?

P4L27: Comma after variability

P5L7: no comma after troposphere

P5L8: comma after consequence

*P5L35: Can you discuss the cause of the zonal differences? It looks to me like the monsoon pattern of southerly winds. Also, this discussion seems better suited to the next section on the seasonal variability.

P6L22: No comma after tropics

P6L23: "... ITCZ, see surface winds" awkward phrasing, comma splice

P8L6: maxima

*P8L25-...: Why is the analysis of one El Nino and one La Nina event preferable to examining composites of high/low ONI? It seems that with a composite you could be more clear about mechanisms.

P9L10: "above" is not necessary

P9L12: "during El Nino year" is either missing an article or year should be plural.

P10L20: need "of" after "deviation"

*P10L29: Using only one model does not seem to be a concern, since this is an SD run. The use of an SD run for transport is potentially problematic, however, and a brief discussion of how close the model is to conserving mass would help alleviate my concerns.

P11L4: no comma after gases

---

## Referee Comment (RC2) · Anonymous Referee #2 · 21 Feb 2018

In this study, the authors analyze the seasonal and interannual variability of transport times from northern hemisphere midlatitudes to the southern hemisphere for 3 different idealized age tracers emitted over North Hemisphere midlatitudes (one for mean age, and two decay tracers with 5- and 50-day decay times). The authors relate seasonal variability to (largely) to the seasonal migration of the ITCZ and interannual variability (largely) to ENSO, but point out some differences among the tracers. For example, the authors note that the largest variability for the mean age tracer is close to the location of the ITCZ, while the decay tracers have peak variability south of the ITCZ.

Overall, the results of the study are broadly consistent with findings of previous work regarding seasonal and interannual variability of tracer transport, and the explanations are qualitatively plausible. I do think, however, that the study could benefit from more

detailed discussion of the interplay of tracer transport and dynamics and convection.

For the discussion of interannual variability (ENSO) in particular, the authors could do much more. For example, prior work on the SPCZ-ENSO relationship points to the axis of the SPCZ "diagonal" shifting generally northeastward during El Niño and southwestward during La Niña (see, for example, Vincent et al. 2011; reference appended below). The authors may want to consider placing their results in the context of such spatial displacements of the SPCZ. More generally, I wonder about the relative role of changes in intensity of convection are relative to changes in its location (as discussed in a two-box model interhemispheric exchange time in Lintner et al. 2004)?

It may also be worth noting that the 1997-1998 El Niño event represented what Cai et al. (2012) have described as a "zonal SPCZ" event, with the SPCZ and eastern Pacific ITCZ effectively merging into a single convection zone near the equator. During other El Niño years, the SPCZ does not experience such an extreme response to ENSO forcing. (Whether zonal SPCZs occur appears to be tied to the flavor of ENSO forcing, as these events are more common during so-called "eastern Pacific El Niños" relative to "central Pacific El Niños".)

Given the consideration of 5-day and 50-day loss tracers, it also seems that performing some analysis with respect to intraseasonal variability, especially the Madden Julian Oscillation (MJO), could be of value.

Other Comments P1, L1: "a ideal age"→"an ideal age"

P1, L18: subject/verb agreement: "majority...have used". Suggest changing "The majority of previous studies" to "Most previous studies"

P1, L21: "into single parameter"→"into a single parameter"

P1, L22: "model-data"→"model-observation"

P2, L11: "from NH surface"→"from the NH surface"

[Figure]

P2, L13: It may be helpful to include a brief description of the TTD, for readers who may not be familiar with what this is.

P4, L1: "time T is much larger"→"time T much larger"

P5, L4: subject/verb agreement: "spatial distribution. . .are"→"spatial distribution. . .is"

P5, Last Paragraph: I think it would be worthwhile to develop a bit more in the way of mechanistic explanation for the zonal variations of age in the tropics. For example, for the relatively high values over the northern Indian Ocean in summer, presumably this is related to the South Asian monsoon, which (relative to winter) has the "ITCZ" located far to the north and relatively strong cross-equatorial flow, particularly over the western portion of the Indian Ocean (with the Findlater/Somali jet). This does seem to be touched on later.

P6, L23-24: The part of the sentence "see surface winds. . ." should probably be enclosed with parentheses.

P7, L1: "round"→"around"

P7, L1: I suggest replacing "where" with "while"

P7, L2: remove "moderate to"

P7, L21: what exactly does "more compact" mean here?

P8, L17: "with both occurring either side"→"with both occurring on either side"

P8, L29: subject/verb agreement: "precipitation. . .that extend. . ."→" precipitation. . .that extends. . ."

P9, L9: "in northern tropical"→"in the northern tropical"

P9, L12: "during El Niño year"→"during El Niño years"

P9, L20: "correlations either side"→"correlations on either side"

P9, L23: subject/verb agreement: "characteristics . . .is. . ."→"characteristics. . .are. . ."

P10, L28: "with more strongly influenced by the trail of the TTD"→"which is more strongly influenced by the tail of the TTD"

P11, L3: subject/verb agreement: "variability. . .suggest. . ."→"variability. . .suggests. . ."

Figure 1: The use of the same color scale with different ranges of age of air makes the direct comparison across the panels challenging. While a single scale over the full range is probably not optimal, perhaps the authors could highlight a few common contours across the panels. Also, including a DJF - JJA difference plot could be useful for discussing contrasts between these two seasons.

Figure 2, caption: "approximation location"→"approximate location"

Figure 4, caption: spelling: "variatiability"→"variability"

Figure 4 also seems ripe for further discussion. For example, the structure of the standard deviation for the 5-day loss tracer in subtropical to mid-latitudes of the South Hemisphere exhibits relatively high variability co-located with not only the SPCZ but also the South Atlantic Convergence Zone and the South Indian Convergence Zone. While I realize that this might be beyond the scope of the present study, I'd be curious to see how the tracers reflect observed synoptic-scale interactions in these convection zones (see, e.g., Matthews 2012 or Niznik and Lintner 2013).

References: Cai, W., et al., 2012: More extreme swings of the South Pacific Convergence Zone due to greenhouse warming, Nature, 488, 365–369, doi:10.1038/nature11358.

Matthews, A. J., 2012: A multiscale framework for the origin and variability of the South Pacific Convergence Zone. Q. J. Roy. Meteor. Soc., 138, 1165-1178, doi: 10.1002/qj.1870.

Niznik, M. J. and B. R. Lintner, 2013: Circulation, Moisture, and Precipitation Relationships along the South Pacific Convergence Zone in Reanalyses and CMIP5 models. J. Clim., 26, 10174–10192, doi:10.1175/JCLI-D-13-00263.1.

Vincent, E. M., M. Lengaigne, C. E. Menkes, N. C. Jourdain, P. Marchesiello, and G. Madec, 2011: Interannual variability of the South Pacific Convergence Zone and implications for tropical cyclone genesis, Clim. Dyn., 36, 1881–1896.

---

## Author Comment (AC1) · 4 Apr 2018

We thank the reviewers for their helpful comments. Our responses to specific reviewer's comments are given below.

**ANONYMOUS REFEREE 1**

The authors have done several new things in this study, and the paper would be improved by highlighting the novel aspects (interannual variability, robust seasonal analysis, comparison with data) of the work. Generally, more discussion of mechanisms, why the results matter, and improved framing would take this paper from being a simple advance in the examination of inter-hemispheric transport to being a more interesting and useful contribution.

[Figure]

**Response**: We thank the reviewer for their suggestions, and as described below we have revised the manuscript to provide improved justification and more discussion of mechanisms and links to meteorology.

Introduction: The previous literature on interhemispheric transport is treated well. It is not clear from the discussion why neglecting the seasonal and interannual variability in inter-hemispheric transport matters, and why it is important to consider both shorter lived and longer lived trace gases. . . . A stronger connection to the chemistry would go a long way for motivating the rest of the study.

**Response**: We have including the following paragraph in the Introduction that follows the reviewers comments: "However, understanding the temporal variability of the transport is important for understanding and interpreting the observed temporal variations in tracer concentrations, and determining the relative role of changes in transport, emissions, sinks, and chemistry for different species. For example, observations of methyl chloroform, or other species with reaction with OH as their primary sink, can be used to infer the abundance of OH (Krol and Lelieveld 2003, Prinn et al 2005, Montzka et al 2011, Liang et al 2017), and knowledge of the seasonal and interannual variability of the transport is required to isolate similar variability in the OH abundance. Similarly, knowledge of the interannual variability of transport from the NH is required for estimates of the variability in emissions or sinks (ocean uptake) of $CO_2$ from measurements of $CO_2$ in the SH (e.g., Francey and Frederiksen 2016)."

In each section, explain what we expect to see. The ITCZ is known to drive interhemispheric transport and ENSO is known to change tropical variability in DJF, so the results are not surprising. Unfortunately, the paper currently reads as a methodical discussion of plots. In my opinion, the story of this paper would be more compelling if it were presented as "this is what we expect to see", "let's check–this is what we see", and "these are interesting details".

**Response**: We have revised the discussion to address the reviewers concerns. In

some places we can start with "this is what we expect", but in other sections we feel it is better to look at general features before focusing on a particular aspect (e.g. while ENSO is a major contributor to interannual variability it is not the only issue and we prefer not to start with a discussion of ENSO).

For the ENSO discussion, an explanation of why the weaker Walker circulation would lead to the pattern of age difference would be nice. The comparison with the wind anomalies in Wang and Fiedler (2006) in Fig. 2c definitely makes Fig. 9a seem reasonable. The current discussion based on the case study comparison of one El Nino and one La Nina year is less convincing than a more general discussion would be. For example, if the variations in deep convection in the SPCZ region are the norm for La Nina years, a citation here would be helpful. For JJA interannual variability, it is not as obvious what mechanisms are at play.

**Response**: We have replaced the case study of ENSO with composites, including those from two other simulations. The discussion now focuses on the general features rather individual features that may be only in a single event.

Perhaps the variability is related to changes in the monsoon and therefore the phasing of the MJO (the strong variability in age and the mean age contours are both apparently coincident with the Somali jet, and the authors discuss the ascent in this region). Since ENSO is the primary signal of interannual climate variability, I would also think that checking whether JJA age is correlated with the ONI would be worthwhile, just in case. I do not expect the authors to do any extensive calculations for these dynamical connections rather, I ask for a discussion of the transport in the context of the tropical dynamics. I think a different, longer model run would be necessary to get at the details here, and that is beyond the scope of this paper.

**Response**: We agree that phasing of the MJO may be playing a role, but we only have monthly-mean output which limits our ability to analyze the MJO. We have however, include a figure and discussion of the ENSO influence during JJA.

I find the comparison to observations makes this paper more valuable; this should be emphasized in the introduction, and the discussion should be expanded (p. 7).

**Response**: We have expanded the discussion on page 7 a little, but there is not a lot more we can say given the limited data, and have included the following in the Conclusions: "Trace gas observations from surface stations provide support for these model results: The SF6 age derived from tropical measurements varies seasonally with the latitude of the ITCZ in a similar manner to the simulated ideal age, and lower concentrations of tracers with NH sources are observed at the America Samoa station during El Nino years (consistent with slower transport)."

The point that is made in this conclusion is that by examining a bunch of CCMI models, the robustness of these relationships could be tested. I would contend that we fully expect the gradients of age to be tightly coupled to convection and that Orbe's work has shown as much–these relationships don't really need to be tested. However, differences in these relationships between models might have interesting implications for the impacts of dynamics and convective parameterization on model transport and chemistry and how they differ in the CCMI models.

**Response**: We have removed this paragraph as this was not central to our conclusions, and we agree with the reviewer that robustness doesn't need to be tested.

*P4L16: Why do you use CAM SD? What advantages (and disadvantages) does using the nudged run have for this study?

**Response**: We have included the following in Section 2. "As the CAM-C1SD simulation uses meteorology from reanalyses it has the advantage over free-running simulations (in which the meteorology is generated internally) in that the tracer distributions can then be directly compared with observations for the same period. However, Orbe et al (2017a) have recently shown there is large uncertainty in specified dynamics simulations due to the transport by parameterized convection."

*P5L35: Can you discuss the cause of the zonal differences? It looks to me like the monsoon pattern of southerly winds. Also, this discussion seems better suited to the next section on the seasonal variability.

**Response**: We have included the following discussion of the cause of zonal variations: "These variations in the tracers can again be related to variations in meteorology. In particular, the large seasonal variation over the tropical Indian Oceans is related to seasonal changes in convection and wind direction associated with the South Asian monsoon, i.e., there is deep convection over the equatorial Indian ocean and northerly surface winds in DJF, whereas the deep convection is over the northern subtropics and there are southerly winds in JJA."

*P8L25-. . .: Why is the analysis of one El Nino and one La Nina event preferable to examining composites of high/low ONI? It seems that with a composite you could be more clear about mechanisms.

**Response**: We have replaced the single event analysis with a composite analysis.

Specific edits (content ones have a *, otherwise these are mostly grammar):

**Response**: All grammatical errors corrected.

**ANONYMOUS REFEREE 2**

In this study, the authors analyze the seasonal and interannual variability of transport times from northern hemisphere midlatitudes to the southern hemisphere for 3 different idealized age tracers emitted over North Hemisphere midlatitudes (one for mean age, and two decay tracers with 5- and 50-day decay times). . . . Overall, the results of the study are broadly consistent with findings of previous work regarding seasonal and interannual variability of tracer transport, and the explanations are qualitatively plausible. I do think, however, that the study could benefit from more detailed discussion of the interplay of tracer transport and dynamics and convection.

**Response**: We have increased the discussion of transport and dynamics in each of the results sections.

For the discussion of interannual variability (ENSO) in particular, the authors could do much more. For example, prior work on the SPCZ-ENSO relationship points to the axis of the SPCZ "diagonal" shifting generally northeastward during El Nino and south-westward during La Nina (see, for example, Vincent et al. 2011; reference appended below). The authors may want to consider placing their results in the context of such spatial displacements of the SPCZ.

**Response**: This is a good suggestion and we have included the following in the text: "The reverse age-ENSO correlation occurs in the southern tropical Pacific because of interannual variations in the SPCZ. During most winters the SPCZ is orientated diagonally in the north-west to south-east direction, but during some strong El Nino events the SPCZ is shifted north and is more zonally orientated (Vincent et al 2011). During these El Nino years there is less rapid transport of younger air from the NH and older air from the SH high latitudes, and hence older tracer ages, in the south-western tropical Pacific."

More generally, I wonder about the relative role of changes in intensity of convection are relative to changes in its location (as discussed in a two-box model interhemispheric exchange time in Lintner et al. 2004)?

**Response**: We have not done a formal analysis of intensity relative to location (this is we think beyond the scope of this study), but have included mention of this in the revised manuscript.

It may also be worth noting that the 1997-1998 El NinÌČo event represented what Cai et al. (2012) have described as a "zonal SPCZ" event, with the SPCZ and eastern Pacific ITCZ effectively merging into a single convection zone near the equator. During other El NinÌČo years, the SPCZ does not experience such an extreme response to ENSO forcing. (Whether zonal SPCZs occur appears to be tied to the flavor of ENSO

forcing, as these events are more common during so-called "eastern Pacific El Ninos" relative to "central Pacific El Ninos".)

**Response**: We have replaced the single event analysis with a composite analysis, and included the following in the text: "These changes are connected to changes in location of ITCZ over eastern Pacific, the SPCZ, and convection over northern Indian Ocean / south Asia. For example, the region of convection over southern Asia is displaced to the north during El Nino, reducing transport of young northern hemisphere air into the region."

Given the consideration of 5-day and 50-day loss tracers, it also seems that performing some analysis with respect to intraseasonal variability, especially the Madden Julian Oscillation (MJO), could be of value.

**Response**: Unfortunately, we only have monthly mean data which prevents any detailed analysis of the MJO or intraseasonal variations.

P5, Last Paragraph: I think it would be worthwhile to develop a bit more in the way of mechanistic explanation for the zonal variations of age in the tropics. For example, for the relatively high values over the northern Indian Ocean in summer, presumably this is related to the South Asian monsoon, which (relative to winter) has the "ITCZ" located far to the north and relatively strong cross-equatorial flow, particularly over the western portion of the Indian Ocean (with the Findlater/Somali jet). This does seem to be touched on later.

**Response**: We have included following discussion at the end of the Section: "These variations in the tracers can again be related to variations in meteorology. In particular, the large seasonal variation over the tropical Indian Oceans is related to seasonal changes in convection and wind direction associated with the South Asian monsoon, i.e., there is deep convection over the equatorial Indian ocean and northerly surface winds in DJF, whereas the deep convection is over the northern subtropics and there are southerly winds in JJA."

[Figure]

Figure 4 also seems ripe for further discussion. For example, the structure of the standard deviation for the 5-day loss tracer in subtropical to mid-latitudes of the South Hemisphere exhibits relatively high variability co-located with not only the SPCZ but also the South Atlantic Convergence Zone and the South Indian Convergence Zone. While I realize that this might be beyond the scope of the present study, I'd be curious to see how the tracers reflect observed synoptic-scale interactions in these convection zones (see, e.g., Matthews 2012 or Niznik and Lintner 2013).

**Response**: We have included more discussion of tracers and SPCZ variability (see below), but, as with the MJO, we cannot examine synoptic-scale interactions with only monthly-mean fields. "Seasonality in surface convergence also contributed to the region of enhanced seasonal variability in the subtropical western south Pacific. The South Pacific Convergence Zone (SPCZ) lies within this region, and the orientation and intensity of the SPCZ varies on synoptic through to interannual time scales (e.g., Matthews 2012, Niznik and Lintner 2013). This variability in the SPCZ then results in variability in tracer ages, e.g., when the SPCZ is shifted to the north-east from its climatological there is less rapid transport from the NH and more from SH middle-latitudes, resulting in older ages."

Other Comments

**Response**: All other comments (e.g. grammatical errors) corrected.